# Interactive Games and Plays in Teaching Physics and Astronomy

Zhuldyzay Akimkhanova [1], Kunduz Turekhanova [1] and Grzegorz P. Karwasz [2],*

1 Faculty of Physics and Technology, Al-Farabi Kazakh National University, Almaty 050040, Kazakhstan
2 Didactics of Physics Division, Institute of Physics, Astronomy and Applied Informatics, Nicolaus Copernicus University, 87100 Toruń, Poland
* Correspondence: karwasz@fizyka.umk.pl

**Abstract:** Physics is a difficult subject in which to trigger interest in pupils, particularly in junior high school classes: this reflects in the results of maturity exams. Therefore, teachers, educators (and authorities deciding on CV contents) should search for new efficient methods, techniques, and contents corresponding to particular topics in physics. What can be done at the level of a single university (even if big) or school is to enrich standard lessons with new elements and observe how the responses of pupils change. In this article, the results of an implementation of interactive plays and games are presented for enhancing pupils' interest and rate of understanding in physics, astronomy, and engineering. The games were designed by authors and the contents were developed jointly at Nicolaus Copernicus University, Poland and al-Farabi Kazakh National University, Kazakhstan. Implementation was carried out both in schools (a secondary school in Almaty and one primary school in Torun) and in extra-school (secondary school students in Almaty, elementary and secondary school students in Torun, during university-based activities) environments. A preliminary analysis of the didactical efficiency is given. We observed a positive reception of the majority of the didactical means that we proposed. These observations will serve us for further (and possibly permanent) enrichment of forms and contents of teaching physics and astronomy.

**Keywords:** education; pedagogy; role-playing games; didactic games; teaching physics

## 1. Introduction

Education plays a majority role in the development of any country and every society understands that quality education is the key to a prosperous future. The Faculty of Physics and Technology of al-Farabi Kazakh National University (KazNU) conducts different research works related to educational processes, which help to establish the principle of multiple choice of adequate forms, methods, and technologies of education [1–3]. In particular, in our previous communications born in collaboration between Nicolaus Copernicus University (NCU) and KazNU, we gave examples of the use of simple interactive experiments [4], interactive lectures [5], and computer-controlled experiments [6]. Elements of physical theatre in teaching physics were discussed at the level of secondary schools. The teaching materials were prepared in collaboration between the two universities and the research was conducted at schools in Almaty, the ex-capital of Kazakhstan.

The game method is one type of practical educational instrument that requires a teacher to be able to search for interdisciplinary solutions and use their imagination. On the students' side, the method aims at developing a vast range of cognitive processes (perception, attention, memory, thinking skills). It is particularly useful in consolidating the knowledge acquired previously in the classroom [7]. The application of games is an effective way of teaching physics to schoolchildren because this discipline explains the nature and ideas of the world around us. In addition, it is the basis of scientific and technical knowledge, which can be applied by students in real life cases. Additionally, the game technique has very deep roots: from the earliest period of training, education uses various

plays and games. Even Jean Piaget, the founder of cognitive psychology, discussed the effectiveness and importance of learning through games [8].

Nowadays, games in education are usually understood as computer-based activities (see other papers in this thematic issue). Knowing this, we came back to the "real" world, in which the games were personal interactions. In particular, role-playing games are widely used in the systems of education, such as teaching specific subjects [9–12], learning foreign languages [13–16], and mastering future professions [17,18]. The usage of games in learning illustrates that this method is very easy; it contributes to students' comprehension and motivates them to improve their knowledge [19–21]. During a game, students can recognize themselves in different roles, observe their activity in cooperation [22], develop their memory and critical skills [23], learn to work independently, enrich their vocabulary, and learn to communicate [24].

In this paper, we present ideas of some interactive games that are constructed with "real" physical objects (or their models). Further, these games rise from an interaction between the teacher and students: both parts play an active, creative role. This general methodology, i.e., using real objects and constructing the knowledge together with students, to a great extent in a spontaneous manner, we called neo-realism and hyper-constructivism (see [25]).

As explained by Giliberti, in physics, interactive role-playing and physical theatre enhance first of all the interest of students and also that of the wider public in this subject, which is one of the most "tedious" to teach in school. However, even if "most students consider physics as an important resource for society, they usually consider it more linked to technology than to general culture" [26].

The direct motivation for our actions in introducing elements of interactive teaching into physics is the generally low level of the social perception of this subject (see detailed discussion in the previous paper [25]). Moreover, despite the development of new technologies in education, the results of the maturity exams in physics, say in Poland, are really disastrous (see Figure 1a). The majority of students (out of 18.9 thousand) did not obtain a sufficient score (30 points on the 100 points scale): the median was 32% and the dominant result was 7% (!). Note that this is not the case for other subjects (see Figure 1b for the results from the same (2022) year in humanities). The maturity exam scores in Polish literature and language can be summarized as "no student left behind"—a double Gaussian distribution, with only a small part below 30 points.

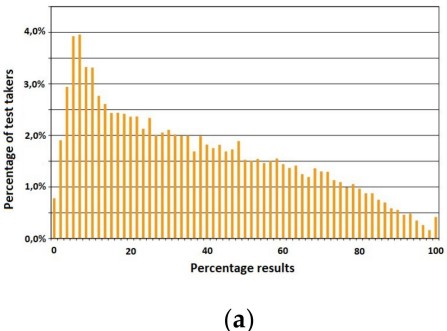

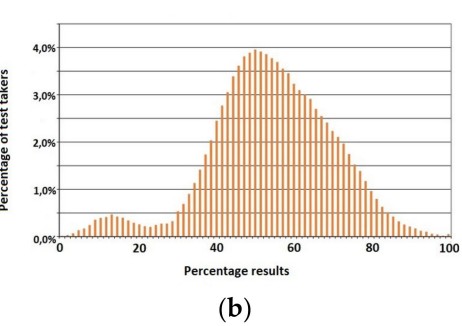

(**a**)                                                    (**b**)

**Figure 1.** Why teaching physics requires new approaches. Results of the maturity exam in Poland in 2022. (**a**) Physics. (**b**) Polish language and literature. The "sufficiency" threshold in Poland is 30%. Source: Report of Polish Ministry of Education [27]. Reproduced with permission.

The reasons for the very poor performance of secondary school students in physics may be various: the mismatch between the school programs and the methods of teaching, a wrong choice of the exam questions by the central committee, or the low level of preparation of teachers. As university researchers, we have limited influence on these "central" factors. What we can do is propose new ways of presenting physics (astronomy, engineering) to trigger the interest of students. This was the main objective of the actions we undertook

by introducing interactive games, interactive plays, and interactive competitions. As will be shown in the evaluation, there was a greater increase in the interest of students than in their formal level in tests.

Our activity introduces elements of "fun" into teaching. Obviously, this is not a totally new approach. In recent years, many authors have proposed similar ideas in volumes under titles such as *Conceptual Physics* [28] (in Poland entitled *Physics Around Us*), *Physical Circus* [29], or even *Manga Physics* [30]. Although these books were published by leading editors in science education, their influence on teachers' methods seems null: we do not find such innovative content or ways of explaining in the school textbooks available on the market. Our impression is that these innovative books remain a kind of "desirable" formulation, but they lack practical implementation, in particular inside the school curricula. In fact, as we (GK) explained previously [31], "fun" is not enough: this is only the gateway for didactical actions and they must be completed by triggering the scientific interest of students. Gilberti [32] stated it in a complementary way: the physical theatre must go outside the academy and beyond mere divulgation.

An infinity of approaches is under testing to enhance the efficiency of teaching physics and to raise the level of its social perception. The methodology presented here is similar in approach (but not identical) to the narrative method of teaching physics [33,34], teaching by the theater [26,32], teaching in science centers [31,35], and so on.

Obviously, the very basis of didactical activities should be a more general, i.e., theoretical analysis. As stated by Jerome Bruner, one of the founders of cognitivism, the search for learning and teaching theory is a permanent, never-ending process [36]. Out of the five most frequently invoked contemporary learning theories (behaviorism, cognitivism, constructivism, humanism, connectivism [37–39], we (GK) lean towards constructivism: teaching and learning via an interactive process, autonomously projected by the teacher and involving the whole school/extra-school group, and based on their already acquired notions (see [25,31] for details). However, the activities and approaches we present here are somewhat different: (interactive) games and plays must be designed in advance, in order to assure clear rules and scheduled scenarios. Therefore, our actions discussed here are based, somewhat surprisingly, on behaviorism. The stimulus is the game: its sequence and the interaction with other "players"; and the "outcomes" from the pupils are their reactions (to a great extent predictable). However, our very inclination towards new methods derives from cognitivism. As stated by Piero Crispiani [40]: "The cognitive didactic [method] is interactive, reads and re-reads, deepens, requires arriving to the nucleus, revealing resources, multiplying cognitive styles, repeating questions via pushing and pulling, interpreting, permanently verbalising the previous knowledge according to a rising spiral of learning, as the sign of the logic of quality" (p. 19). Therefore, even if we stay within the range of our common previous various actions (see [4–6]), i.e., within the constructivist framework, the present methodology is closer, as said, to behaviorism—a stimulus and the outcome: a "round" of the game and pupils' reactions.

In this paper, we show also that interactive teaching may introduce forms typical to humanities, so we show physics as a part of the more universal culture: we mix physics, astronomy, and space flights with the history of science, arts, and philosophy.

The practical research question was: Which forms and methods trigger the interest of students (mainly in the lower secondary schools, according to previous Polish and present Kazakh standards, i.e., in the age 12–14 years), to stimulate their interest in further studies of physics astronomy and engineering? We evaluated the immediate impact (collecting impressions of pupils and by formal tests) and the long-term impact—by following university choices of pupils participating in interactive forms of teaching.

In the theoretical part (see Methods), we describe the requisites that should be included in the process of designing educational interactive games and play. In the practical part, we discuss different, interactive forms of plays and games that our two groups developed for teaching physics and astronomy.

Three forms were designed, implemented, and underwent a preliminary evaluation in the period January-February 2022, in view of this Special Issue on designing educational games: (1) "A court of justice on the electrical resistance", (2) "A duel on the electricity", both tested in the 8th form of the elementary school in Almaty, Kazakhstan; and (3) a didactical "tunnel" on astronomy designed for the same school. To give a more complete choice of forms and contents, we discuss also two other plays that we have been developing for several years: (4) a competition on astronomy in Almaty for secondary schools (15–16 year olds), and (5) interactive lectures with role-playing experiments on astronomy and space flights run in Torun, for pupils aged 8–16 years.

The general aim of designing our games (and preliminary evaluation of their efficiency) was to find forms and contents that would: (i) interest pupils and students, (ii) improve their understanding of problems that are particularly difficult in their current school curricula, (iii) and possibly exercise a long-term impact. Obviously, the grade of fulfilling and the interdependence between these aims depend on the particular applications we have developed, as listed in the previous paragraph. The detailed objectives of the study are given at the end of the section "Materials and Methods".

We show how a proper choice of form and contents makes physics interesting in different environments, first of all in lower secondary school. Our collaboration shows also that the same subjects—the discoveries of Copernicus, Galileo, and Newton—can be easily translated into other national content.

## 2. Materials and Methods

### 2.1. Role-Playing Games in Teaching

Didactical methods based on games and interactive plays help to develop students' minds and increase their interest in the lesson. It is known that even children with low interest and lack of effort participate in the lessons mixed with game elements with great enthusiasm and energy. The conditions of organization and conduction of didactic games must contain both clear didactical goals as well as emotionally involve the students; the whole conceptual scheme [41] can be described in the following picture (see Figure 2). As seen from this figure, the keywords in designing educational games are: (i) a clear purpose of the lesson, (ii) simplicity in explanations, (iii) the allowance for creativity, (iv) space for pupils' emotions.

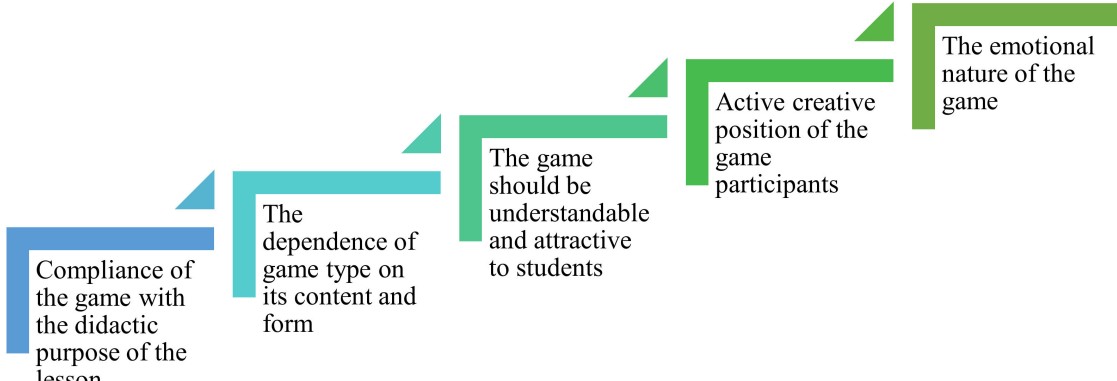

**Figure 2.** Conditions for designing game technologies to be applied in education. Scheme by the authors (ZhA).

A necessary condition for developing an educational game (or an interactive play) is a process of careful planning. One has to define thoroughly: (i) the didactical goal (notions to be taught or repeated), (ii) the means (i.e., objects to be used—physical, textual, hypertextual), (iii) role-playing scenarios. We list these methodological, sequential requirements in the following scheme (Figure 3).

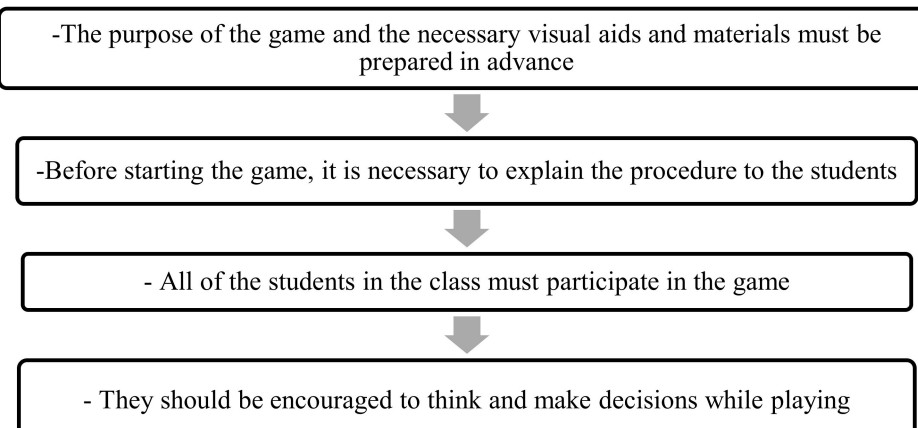

**Figure 3.** Requirements for the organization of didactic games. Scheme by ZhA.

In the above scheme, we stress the importance of preparing the very concept of the game: the didactical goal that is formulated by the teacher, in advance. From this goal, and keeping in mind the level of perception of the addressees, visual aids and materials are defined.

Let us distinguish two variations of our method. In the educational game to be executed as a part of the didactical curriculum in school, we present all aids and materials immediately at the beginning, and to the whole class. In the form that we call an educational play, which we develop for complementary, extra-school teaching, we mix elements of a game and of interactive theater: the whole class participates, but the experiments/performance is executed by the pupils' selected ad hoc, who act as performers (see Figure 4). The sequence of experiments is somewhat of a surprise for the class; further, the scenarios are flexible, adapting their order to the perception of the public: the play is really interactive.

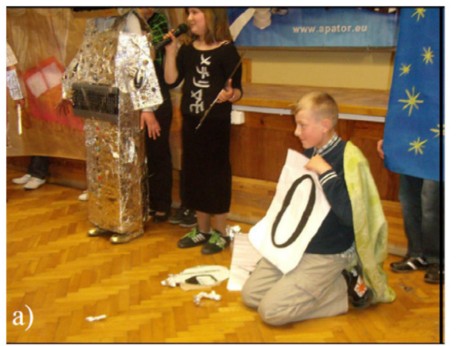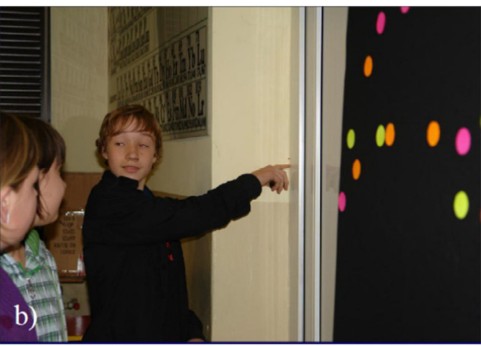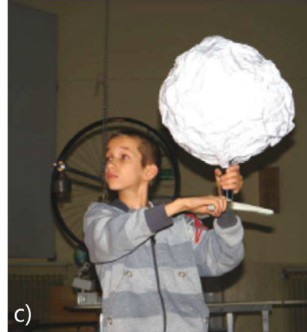

**Figure 4.** Three examples of interactive games for school children that were developed at NCU. (**a**) Theatrical performance by elementary school children: *Robots' Fairy Tales* by Polish s-f writer Stanislaw Lem (2008). (**b**) Searching for planets in an interactive play on the Solar System. (**c**) Galileo's discovery of shadows of craters on the Moon. Author of scenarios GK, source [31], reproduced with permission.

Further, in the first class of games, those tried during lessons, we define two variations. The first one is "role-playing": all actions of a student are determined by the role that is assigned to them in advance and which they perform in this game. In preparation for the game, all participants need to familiarize themselves with the purpose and rules of the game; in a colloquial meaning, this form resembles a "social game". A different form that we experimented with is a "competition", in which some pupils represent the whole class and test their knowledge with another group.

In Figure 5, we show the classification of the role-playing game in teaching. Obviously, this is only a part of a whole variety of forms that individual teachers can adopt/invent.

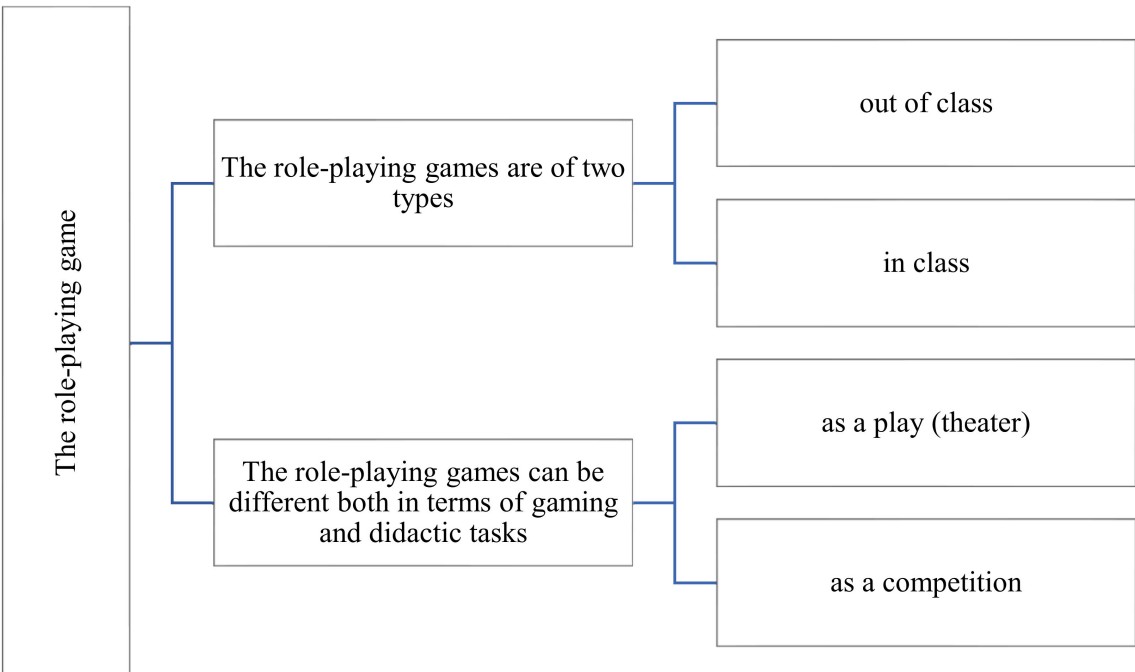

**Figure 5.** Basic characteristics of role-playing games, according to their most popular forms. Scheme by the authors (ZhA).

As explained in the vast literature on the subject, the usage of role-playing games plays a majority role in the explanation of complex processes in physics, chemistry, and natural sciences, as it requires basic knowledge of these disciplines [42–44]. In particular, the game technique has been used in teaching physics for a long time [43,44]. Recently, role-playing games have been applied in combination with new technologies: a positive effect on the teaching process has been verified [45–47]. Successfully organized role-playing games have an important place as an additional element in education [48]. Much information about the usage of role-playing games in the teaching of physics is given on the internet [49], and several research works consider that this method is useful also for astronomy [50,51].

The role-playing game is one of the effective interactive methods. Most scientists who conduct research in the field of pedagogy consider the learning process through games, which are active and impressive for students. Moreover, it is generally acknowledged that the process of educating through activity is the most efficient way of gaining knowledge. It is known that at the first moment, the impression of one's own experience is kept in the memory of any person. In addition, the role-play method develops the selfish skills of students, such as considering the problem from their point of view and creating different types of solutions [52–54].

In this paper, we explore the potential of different forms of interactive games and activities in teaching about the Solar System and astronomy in general, as well as electricity and electrical circuits. The broad didactical targets are pupils of primary and secondary schools (aged 8–16), both in Poland and Kazakhstan. Note that the present educational systems in the two countries are similar now: 8 (9) years of primary school and 4 (3) years of secondary in Poland (and Kazakhstan), respectively. Details of the target groups are given with a description of the specific actions. Generally, for actions in schools (role-playing games), the target groups are single classes (20–30 students); for extra-scholar actions (interactive plays on astronomy) we gather 200–250 students of various ages at every event.

*2.2. Interactive Solar System*

Astronomy, together with mathematics, is one of the first sciences formulated by man. The channel leading from the pharaoh's funeral chamber in the Great Pyramid of Cheops pointed towards the alpha-Thuban star that was the polar star 5.5 thousand years ago (it is not anymore, due to the precession of Earth's axis). Megalithic "temples" of Ħaġar Kim in Malta, dated 5.5 thousand years ago, certainly served astronomical (i.e., calendar) purposes. The most ancient compendium of astronomical knowledge, the so-called MUL.APIN clay table from Mesopotamia, is dated to 700 BC. Astronomy is therefore also a gateway that can be used to arouse students' interests and thus introduce them to science studies.

A simple form, easy to develop and pretty engaging, we call a didactic "tunnel": this is a sequence of exhibits (objects, models, pictures, posters) that are conceived to capture the attention of a visitor. Recently (in 2022, i.e., shortly after the COVID-19 restrictions were lifted) we (ZhA) constructed such a tunnel in the subject of astronomy (the Solar System in particular) for the secondary school (15–17 year olds) in Almaty as a part of the "Ten days for science" initiative (see Figure 6). The tunnel gathered pictures, posters, and interactive objects, all prepared by students, under the guidance of the teacher (ZhA).

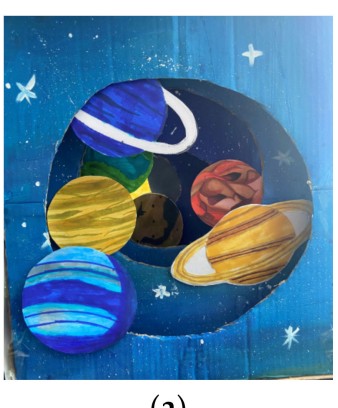 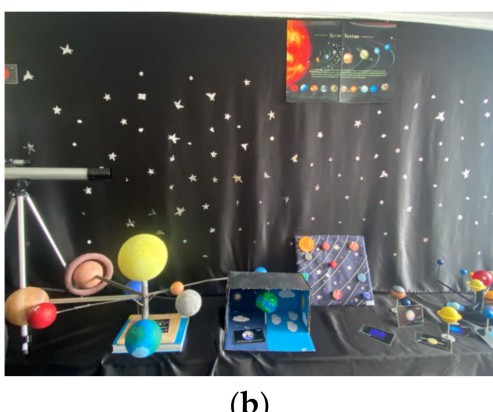 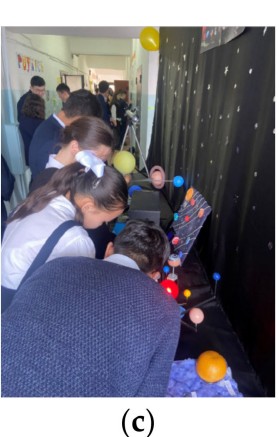

(**a**) (**b**) (**c**)

**Figure 6.** A didactical tunnel as a form of interactive playing with objects: "Ten days on science" at Abai Republican special secondary boarding school in Almaty (January 2023). (**a**) Students' vision of the Solar System. (**b**) Interactive objects to play with. (**c**) The public is deeply involved in the interactive visiting. The idea, implementation, and photos are by the authors (ZhA and KT).

As you see from Figure 6c, the whole group of visitors tried sequentially all interactive experiments and acquired knowledge autonomously, step-by-step, but in the order rigorously planned by the author.

Another way of constructing interactive narration on astronomy is to follow the historical development of the discoveries. We (GK) developed originally interactive scenarios on astronomy in 2009, at the 500th anniversary of Galileo's astronomical discoveries (see Figure 4b,c). Obviously, preparing them in the town of Copernicus, we started with the heliocentric theory, but we did not treat it as a "theory" or as a "model": this is a physical reality, and interactive teaching needs to reproduce real details of the Solar system.

As already mentioned, we apply a "hyper-constructivist" (H-C) method of teaching [25]. Briefly, it is a process of interactive discovery, led (in a sublime way) by the teacher, but allowing autonomy of pupils/students. What differentiates the H-C method from the Socrates-like maieutic method is the huge base of notions that contemporary pupils possess from educational books, TV, and the internet. The role of the teacher is just to raise, in the minds of students, correct questions that lead to the given didactical goal. In the H-C interactive games on astronomy, these goals are the essential information on the Solar System: not only that the Earth moves around Sun, but with what velocity, at what distance, and in which direction. The scenario reproduces these questions step by step.

The first question that we ask the public is in which direction Earth rotates. The answer (given by Copernicus) "from west to east" says little. How can we spot it? In a way it was answered first in Mesopotamia: by vertically inserting a stick into sand and watching in which direction the shadow rotates. Following the second principle of our methodology, neo-realism [25], we must show it. We need a globe, and the sun (a lamp); the shadow of a stick kept vertically on the northern hemisphere rotates "clockwise" (see Figure 7a), or rather vice versa: the arrows of the clock rotate in the direction that the shadow moves, but only in the northern hemisphere. "If clocks were invented in Australia, their arrows would move in the anti-clockwise direction!". Furthermore, according to the N-R principles, we show an anti-clock (Figure 7c) and our photo of a sun clock in Australia, with the "reversed" order of hours (Figure 7b).

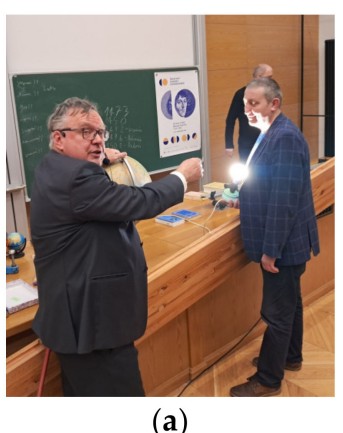
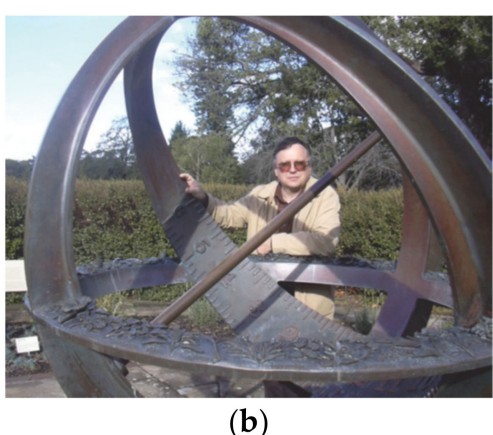
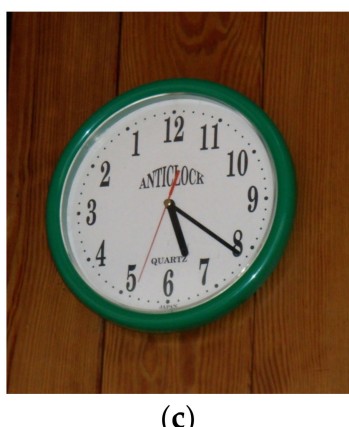

(**a**)          (**b**)          (**c**)

**Figure 7.** "E pur si muove" ("It moves, however," attributed to Galileo)—interactive sequence (objects and images) illustrating the rotation of Earth. (**a**) Constructivist discovery of the relation between Earth's rotation and the direction "clockwise", the interactive play at UMK, 22/02/2023. (**b**) The sun clock (the shadow of the gnomon) moves anticlockwise in Sydney. (**c**) The "anti-clock" on the wall in an NCU lecture hall. Idea and implementation GK, with help in 7a from Andrzej Karbowski, photos Maria Karwasz.

Having explained the rotation of the Earth, we can proceed with explanations of the whole mechanics of the Solar System. "We need three volunteers. You will be Earth, you the Sun, and you the Moon. How long does it take to Earth to make a complete revolution?"— "24 h". "Yes, you are right, or better—almost right. It takes 23 h 56′ for Earth to make a complete rotation as compared to fixed stars. Why? Because in one day it travels also 1/365 of the path around the Sun." The H-C teaching goes step-by-step. So, the next question is: "In which direction does Earth make its revolution around the Sun? The same in which it rotates: anticlockwise; if seen from above the northern hemisphere."

The interactive plays on astronomy, organized for Kopernik's birthday, each year attract numerous visitors. In seven sessions organized on 22 and 23 February 2023, we hosted at NCU premises some 1600 pupils, aged 8–16 years. Only a part of this vast public was asked to give feedback that served us for the validation of the methodology. This limitation was mainly due to the problems of security while organizing such huge (200–250 persons) lectures.

Some of the photos from this interactive play are shown in Figure 8, and the perception of this scenario by pupils of elementary schools in Poland is reported in the chapter Results.

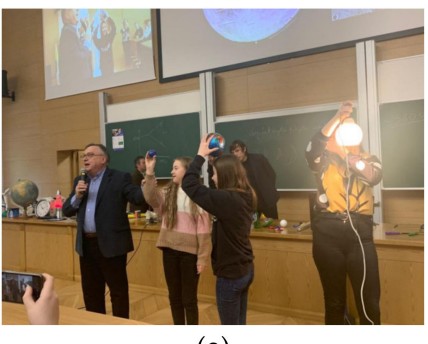 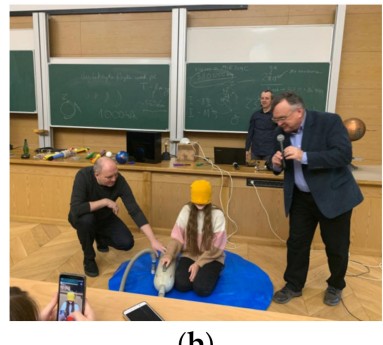 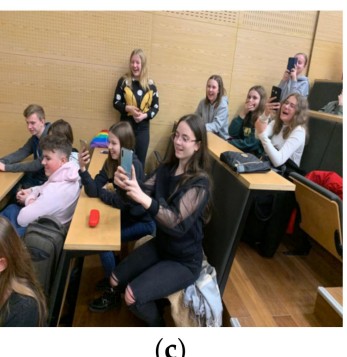

(**a**)  (**b**)  (**c**)

**Figure 8.** "Copernicus, Galileo, Newton, Einstein"—interactive lectures at NCU, 19/02/2020 (birthday of Copernicus). (**a**) "The Solar System"—interactive role-playing: "Earth rotates in 23 h 56 m and revolves around the Sun, the Moon revolves around the Earth. Does it rotate also, and the Sun?" (**b**) "Galileo's relativity principle"—travel on a hovercraft: cold, dark, silent. (**c**) The show brings much fun. The ideas, implementation, and performance GK, helped in 8b by Waldemar Krychowiak. Photos by ZhA.

### 2.3. Game-Based Competition in Astronomy

In competition games, pupils can increase their cognitive activity, develop their interest in studying and, above all, more easily achieve the goals set in the training. Competition games also help students to appreciate themselves, to repeat, and to remember the knowledge acquired in the lesson [. Kazakh schools have recently undergone intensive reforms, both in form and content. Recently, some competitions among students have been held in the form of games; for instance, IYNT (the International Young Naturalists' Tournament), IYPT (the International Young Physicists' Tournament), and M. Abdildin international tournament "Humanity, Earth, Universe" etc. Usually, 50–100 students, divided into squares of 4–5, from different schools enrol in such tournaments.

We have practised the role-playing game with students of the Republic Physics and Mathematics School, in Almaty, in extra-scholastic activities, for several years. These tournaments are conducted as a group competition between the pupils on the ability to solve research problems in the natural sciences, convincingly present their solutions, and defend them in scientific discussions or "fights". The scientific fight is a face-to-face meeting of three (or two) teams, where the solutions to tournament tasks are presented by the teams and are discussed in a jury session. Every team consists usually of 5 members; in the battle, they should change their roles: as a reporter, an opponent, or a reviewer—one member should play a role only one time in the tournament. If these rules are not satisfied, the team could be disciplined and be given a yellow card. The characteristics of each role are given in Figure 9.

Figure 10 shows the competition on the topic of the Moon. The task was: "The apparent size of the Moon perceived by an observer depends on multiple factors. Investigate these factors and their role". Figure 10a shows one team finishing their report and awaiting the assessments from the other teams. An opponent from the other side evaluates the presentation, scientific results, and other skills of a reporter in the form of a conclusion, and their action is presented in Figure 10b. As a result of such tournaments where pupils change their roles in the game, they can form and develop critical, cognitive, and research skills. This tournament was organized outside the classroom, in the city cultural center. Due to its great success, we have repeated it every year (apart from during the COVID-19 period).

**Reviewer**

- Reviewer summarizes and analyzes the discussion between the speaker and the opponent, gives concise feedback to the speaker and the opponent's speech, highlighting positive points and possible flaws; conclusions regarding the discussed issue

**Opponent**

- Opponent makes a speech by giving constructive feedback to the solution, made by the speaker, as well as content and the form of the report

**Jury**

- The jury assesses their preparation for the game and their role-play.
- Time should be followed strictly according to the regulations

**Reporter**

- Reporter gives the team's solution to the problem orally

**Figure 9.** The characteristics of each role in the interactive competition. See text for detailed implementations in the tournament IYNT http://iynt.org/ (accessed on 1 March 2023).

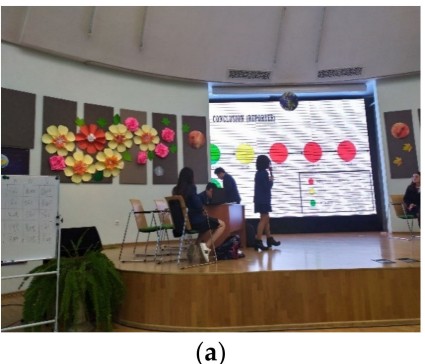
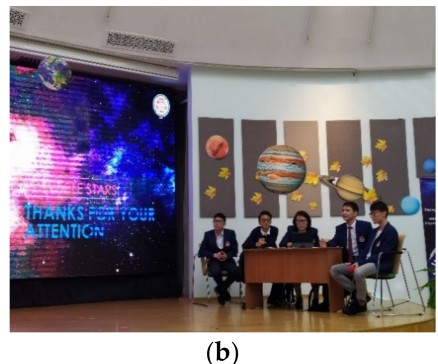

(**a**)                              (**b**)

**Figure 10.** The students of the Republic Physics-Mathematics School, Almaty in the "Astronomy" competition role-playing game, 2019. (**a**) The reporter team finishes presenting on the topic "the Moon" and expects the scores for their work. (**b**) The opponent from the other team assesses their scientific results, the presentation, and expresses an opinion. Photos KT.

### 2.4. Role-Playing on Electricity

The search for efficient teaching on electrical circuits is a vast task. Usually, the flow of electrical charges is illustrated by microscopic pictures or by analogy with liquids [55]. Electrical circuits are also the favorite subject of online simulations [56]. Recently, online methods of teaching electrical circuits have been applied at the university level [57]. However, pupils in secondary schools usually do not show sufficient imagination to understand different elements constituting electrical circuits. To overcome this difficulty, we organized a simulation of a legal court, on "the evil role that the resistance plays in harming the flow of the electrical current."

The role-playing game, called "A court case of the electrical resistance" was based on the last chapter, "Electric Current", of the textbook for students in 8th grade (13–14 years old)

in Kazakh state schools. It was projected, implemented, and tested by the authors (teacher ZhA). Before playing this game, the students were given the theme, the goal, the idea of the game, etc. Separately, we explained the rules of the legal–court game: the necessity to include the prosecutor and the defense and the importance of "peaceful" argumentation, in a consistent and complementary manner, between the parts of the court.

The detailed scenario of the "court of justice", see Figure 11.

(a) Prosecution's witness reports the statement that the electric resistance, in the electrical circuit, is redundant.

(b) A witness called by the defense explains to the prosecutor that resistance is very important for the electrical circuit, the resistance increases when the connection is in series and decreases when the connection is parallel.

(c) Prosecutor: "I believe that the above examples of witnesses clarify the question when in practice it is beneficial to increase the resistance and when it is beneficial to reduce it. Of course, this is relevant to the case, but in principle, it does not solve the issue."

(d) Finally, the registrar of the prosecutor shall read out the decision of the court: "The court finds the resistance innocent!" Now, students link the terms "electric current", and "resistance" associated with this game, which was successful and interesting in their opinion.

In this "court of justice" game we define the individual roles: (1) the prosecutor, (2) the suspect (i.e., the electrical resistance), (3) the resistor's defender, (4) the judge, and (5) the secretary. The objective of the game is to explain the physical meaning of electrical resistance, and its different roles in serial and parallel connections in the circuit. The aim of the game is to show that we should not rely on a hunch, but seek to understand and test an idea that leads to a deeper cognitive involvement in physics, recalling it as an interesting subject [53].

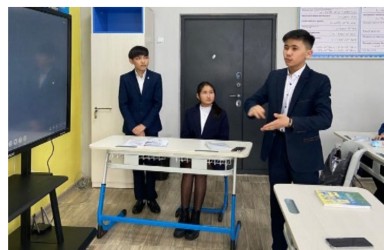 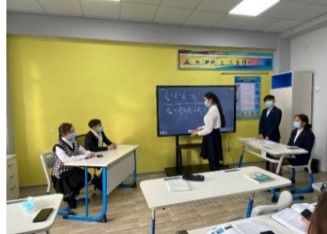 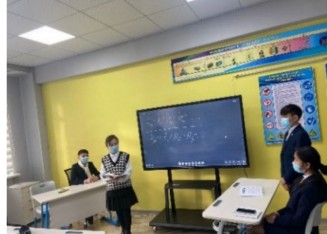

**Figure 11.** "The court case on the electrical resistance"—a role-playing game for secondary school (8th grade). Students of Abai Republican special secondary boarding school for gifted children with extended learning in Kazakh and literature (8G). Idea, implementation, and photos by the author (ZhA), 2022.

We experimented with another form of play on the electricity in the classroom during an 8th grade physics lesson at Abai Republican secondary boarding school. This was conceived as a repetition of knowledge acquired from the chapter on physics, "Electric current". This role-playing game has a form of a competition with the general title "electric current". In contrast to the "court of justice" game, the whole class participates in this competition. The implementation was by the same teacher who led the "court of justice" game (ZhA).

We divided the class into two groups: at the beginning, the teams defended themselves using the answers to the questions. The teacher distributed a card to the captains, on which the questions were written, and the team decided on the answer together (teamwork), Figure 12a. Then, the groups presented the questions, and the teachers (in the center of Figure 12b) let them attack. Each student prepared one question (since it is a duel, the question must be good so that the opponent cannot answer), as shown in Figure 12c. Obviously, there are some questions from the two groups which are identical, and some which are too difficult. After having selected admitted questions, the game goes on as a

sequence of duels in pairs. As we can see from the photos, the whole lesson is more a joyful play than a competition.

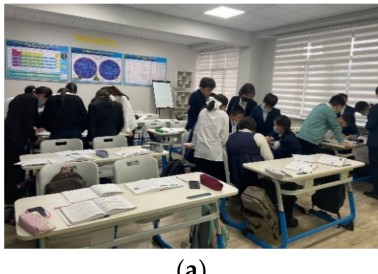 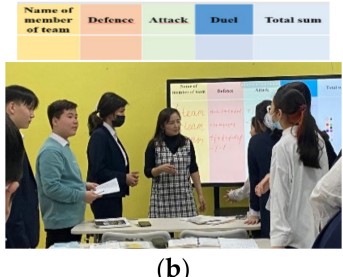 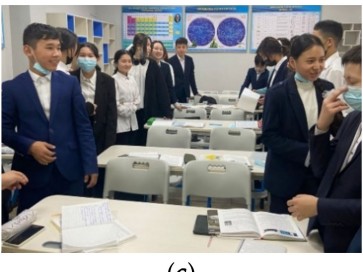

(**a**) (**b**) (**c**)

**Figure 12.** A role-playing game as a competition: "Electric current". Students of Abai Republican special secondary boarding school for gifted children with extended learning in Kazakh and literature (8V). (**a**) The class is divided into groups that individually prepare questions for adversaries. (**b**) The questions are presented and evaluated by the teacher (ZhA, in the center). (**c**) The individual duels in pairs. Almaty, 2022. Concept, implementation, and photos by the author (ZhA).

Questions that each student prepared for opponents in the "duel" section are presented in Table 1.

**Table 1.** The duel between two parts of the class in the competition on electricity: questions prepared by students to opponents. Since these questions are prepared without books and notebooks, they prove that the students have learnt the subject: we give original formulations as invented by students. The didactical experiment run by ZhA, at Abai Republican special school for gifted children, November 2022.

| Team A | Team B |
| --- | --- |
| 1. Does the resistance depend on the electric current? | 1. What is a resistor? |
| 2. Explain Ohm's law? | 2. What is a dielectric? |
| 3. On which quantities does the electric current depend? | 3. Will there be a magnetic field between conductors with the current? Give example. |
| 4. What is an electron? | 4. How does temperature affect resistance? |
| 5. What quantities does the electrical resistance depend on? | 5. Does resistance depend on voltage? |
| 6. What is the physical meaning of resistivity? | 6. What is a charge? |
| 7. Formulate Joule-Lenz's law. | 7. One coulomb is equal to? |
| 8. What are fuses? | 8. How does the length of a conductor depend on resistance? |
| 9. One ampere is equal to? | 9. What is the field strength? |
| 10. Does the electrical resistance depend on temperature? | 10. Formulate Ohm's law? |
| 11. Explain the law of conservation of charge. | 11. Tell us about the parallel connection of conductors. |
| 12. Tell us about the serial connection of conductors? | 12. What is electrolysis? |

*2.5. Research Questions*

We tried five different forms of teaching and triggering students' interest in physics and astronomy, all developed within modern learning theories.

Three research questions were common to these implementations:

1.  Is the use of games in teaching physics unnecessary and/or effective?
2.  Can the game method be applied in any field of physics and/or astronomy?
3.  A well-organized play is fun, but does it affect academic achievement?

The overall objective was to develop an interest in physics and to connect physics with everyday life. In every play, we defined specific aims that were: (a) the physical contents and (b) the social interaction. In the three competitions described, these aims were, respectively:

1.  A court case on electrical resistance:

    (a)    The definition of the resistance and rules of constructing electrical circuits;

    (b)    Playing roles by heart and taking personal positions at the end.

2.    A role-playing game as a competition: "Electric current":

    (a)    Repetition and consolidation of previous themes through competition;

    (b)    An active role (of a teacher vs. a student)—formulating questions and evaluating answers.

3.    Game-based competition in astronomy:

    (a)    Scientific knowledge and competitiveness qualities;

    (b)    Play each out of the five roles, according to the rules of the game.

In all games, students were obliged to show organizational abilities, which are the fundamental condition of constructivism: writing scripts for games, learning according to the individual abilities of each student, and respecting different points of view. In this way we lead students far beyond mere knowledge, towards an active, cognitive performance: physics and astronomy become tools for exercising pedagogical actions.

### 3. Results

Our observation is that in general, the application of any type of game in the classroom expands the worldview of students, developing their thinking intelligence, as well as increasing their activity, interest, and creativity. In particular, we noted that astronomy tournaments embolden and inspire students for independent research and develop students' talents and imagination. Unfortunately, due to the limited number of students in such tournaments, it is difficult to evaluate the didactical outcome, which means that most of the schoolchildren could not take part in such competitions. However, we traced that participation in such events helps our students to have brilliant academic careers: presently most of these pupils study at different top universities in the world, such as Berkeley and Purdue Universities (USA), Hong Kong Baptist University, Korea Advanced Institute of Science and Technology (Daejon, Republic of Korea), or Nazarbayev University (Kazakhstan). Out of 12 these participating students, only two of them study physics (and none of them astronomy); the others enrolled in different engineering faculties (civil, mechanics, materials etc.). This agrees with what we stated in the chapter Methods: astronomy acts more like a key to open students' interest in sciences (and engineering) than as a goal in itself.

The didactical tunnel on astronomy in two days was visited by the whole school, i.e., about 280 students. Electronic feedback was received from 75 students, and all of them showed highly positive results (98.7%): 47 students out of 74 wrote that they had never seen such layouts anywhere.

Here, we list some of their answers to the question: "What did you feel when you first saw the astronomical tunnel?"

-     It is like I travelled into space;
-     There was a feeling that celestial bodies could be touched with hands;
-     I saw the space like in a real life;
-     I seemed to be looking at the world from aside.

The particularly positive feedback we received was from students involved in the preparation of the exhibition. Here, we cite some keywords they used in relation to their work:

-     Creativity;
-     Infinity fantasy;
-     My interest in the study of the world awakened;
-     How many ideas came to my mind;
-     There was an interest in science.

The games performed within school lessons, with whole classes, were easier to evaluate. The lesson on the theme "electricity" was taken by the same teacher (ZhA) in three

classes, 8B, 8V, and 8G, and compared: two classes (8V and 8G, in total 46 students) used di-dactical games, and one class (8B, 23 students) was a control group. In the control group the subject "electricity" was repeated in a traditional way, following the compulsory textbook. The results of the study show that in the classes where the game technology was used, the number of correct answers was significantly better. As far as the combined starting level of all three classes, this was similar (77–79%); the control class reached a level of 83% after the repetition lesson, while the two "game" classes arrived at 87 and 96%, see Figure 13.

**Table 2.** Comparison of quiz results before (left part) and after (right part) applying game teaching. Two experimental classes, in which the interactive role-playing game was used as a refresher lesson, are compared with the control group (8B), in which consolidation was done in the traditional way. The first series of five columns give results before the consolidation lesson. The next five columns give results after the consolidation lesson. Project and testing by authors (ZhA), November 2022.

| Class | No. of Students | Grade | | | | Quality % | Grade | | | | Quality % |
|-------|-----------------|-------|-------|-------|-------|-----------|-------|-------|-------|-------|-----------|
| | | «5» | «4» | «3» | «2» | | «5» | «4» | «3» | «2» | |
| 8 «B» | 23 | 2 | 16 | 5 | - | 78% | 4 | 16 | 4 | - | 83% |
| 8 «V» | 22 | 5 | 12 | 5 | - | 77% | 5 | 16 | 3 | - | 87% |
| 8 «G» | 24 | 5 | 14 | 5 | - | 79% | 9 | 15 | 1 | - | 96% |

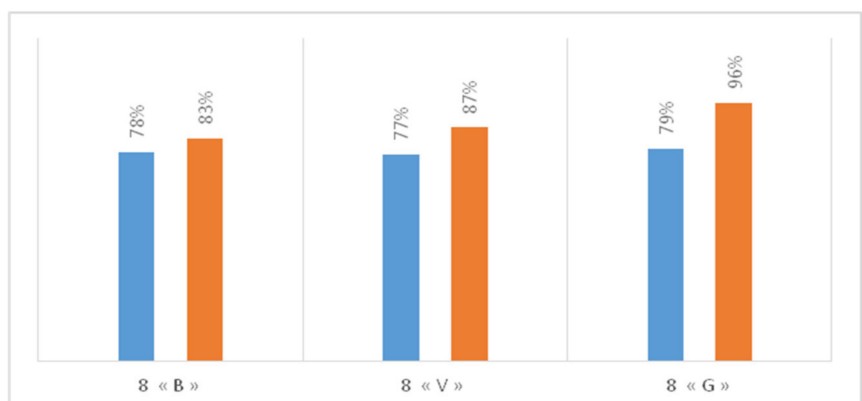

**Figure 13.** Results of quiz on electricity: blue (left) bars—before applying game teaching, red (right) bars—after applying games. "8B" is the control group, "8V" and "8G" are the experimental groups (where the game "electricity court" was tested). The ordinate scale is relative, it reflects the rise of students' final outcome in the two compared cases; see also Table 2. Idea and testing by authors (ZhA).

In detail (see Table 2), before the final (the game or the traditional repetition) lesson, in three classes there were five children with a rating of "3" (i.e., "sufficient"). After the final lesson, in two experimental classes where games were used, the number of "3" scores decreased: in 8V—from 5 to 3 pupils, in 8G—from 5 to 1 student, while in the control class 8B it decreased less—from 5 to 4 pupils. The significant growth was seen in the number of "5" (i.e., excellent) votes: from five to nine in 8G; see Table 2.

Student surveys were conducted to determine the effectiveness of game methods: they involved all 69 students, 2 classes that used the game method (46 students) and 1 class that did not use it (23 students). As we expected, the two groups had different opinions.

As shown in Figure 14a, about half of the 23 students who did not use the game "did not support" the use of game technologies in the teaching process. In contrast, there is shown in Figure 14b that for those who used game technologies, more than 90% of students answered that "games are useful". In the common opinion of this group, the game-based lesson was not only more effective than a standard lesson but first of all also more fun.

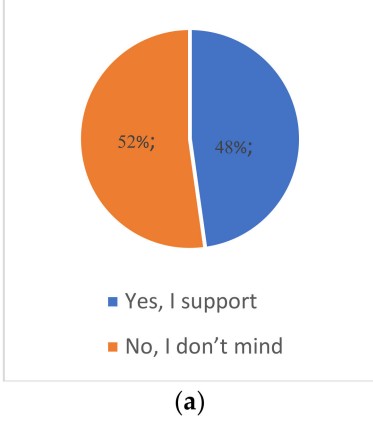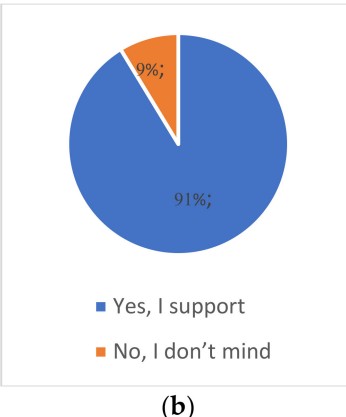

**Figure 14.** Students' feedback on using the game method in the physics teaching process. "Do you think that games are useful as a school lesson?" (**a**) Class 8B (no games trained during the lesson); (**b**) Classes 8V and 8G, which experienced the game-based lesson. Source: the author (ZhA).

Further, we asked students from these three classes the question: "For which subject is it effective to use elements of games?" The control group (8B) gave answers that were rather "stochastic": single votes went to biology, mathematics, physics, geography, economics, and so on; 35% of students in this class answered, "I don't know".

The experimental classes, where we tried the games, expressed a similar spread of opinions, but as many as 14% of students answered that "games are particularly useful in teaching physics" and 24% considered that application of games to all disciplines (both scientific and humanistic) is effective.

The "court of justice" does not involve the whole class, but also in this game, the perception was positive. Students were shocked that physics could be so associated with games, especially role-playing games. For the first time in their lives, they saw role-playing in the form used during a lesson. As seen in Figure 11, they were emotionally involved in their roles.

Our observations from the "court of justice" are:

(1) Rather than solve didactical difficulties directly, using textbooks, it is sometimes better to organize a game that reproduces the notions (and mathematical laws also) we want to construct.

(2) This process of emotionally involving constructing is more important than the very notions that they should immediately remember.

(3) "We know what a resistor is, but after such a game, this knowledge (that the value of electrical resistance differs with a parallel and serial connection) remains in our memory forever!"—one of the participants stated.

Obviously, we are aware that the results are not statistically significant. Our aim was to design innovative, interactive games rather than to check their efficiency in a large (and methodologically refined) manner.

Finally, we evaluated the interactive plays performed at NCU in February 2023. This was achieved indirectly by TV reporters who broadcasted the event. "What did you like in this lecture?" was the question asked by the TV, and these are the answers of 13-year-olds pupils while leaving the lecture hall:

- Many interesting experiments;
- The whole lecture as it was narrated;
- The fact that the professor answered my question.

The question that we posed regarding the interactive experiments was: "Which of these five plays did you like most? Put 1, 2, 3 (or none) asterisks."

(i) Solar clock on the Earth's globe (see Figure 7a);
(ii) Playing with the Solar System (Figure 8a);

(iii)   Sun and Earth in space (Figure 4b);
(iv)   Rotating Earth like a spinning top;
(v)   Travel on a hoover craft (Figure 8b).

We tested three classes from elementary schools (aged 11–13 years) in Torun and the neighborhood and one secondary class from Torun. Results of the evaluation from a sample of 20 students aged 12 years are given in Figure 15a. As seen from the figure, all experiments attracted the pupils' attention. Obviously, the most unpredicted (and therefore funny, like a hovercraft) received three stars, but pupils liked also the interactive play, with the three roles, on the Solar System, that can be easily repeated without any equipment. In the secondary school, students appreciated also less "appealing" but more conceptual plays, like those on the Earth and Sun in space (see Figure 15b).

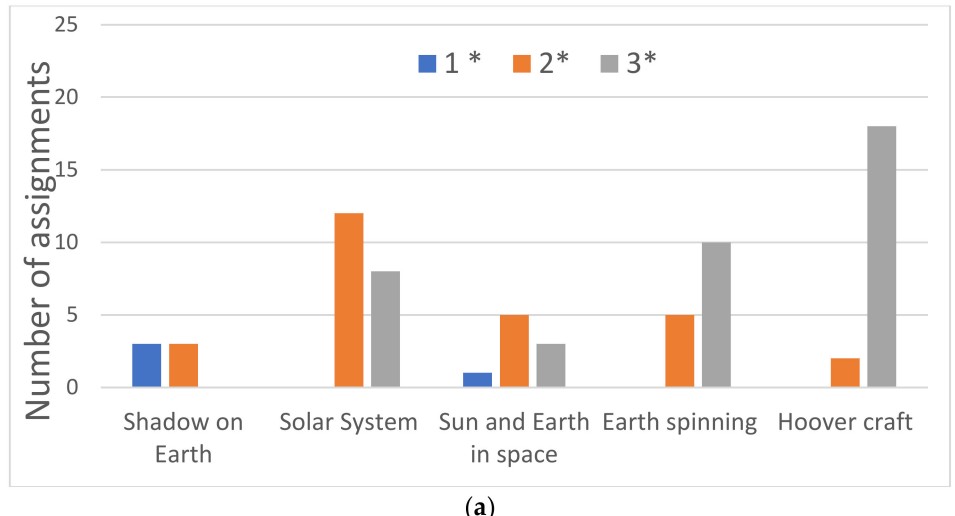

(**a**)

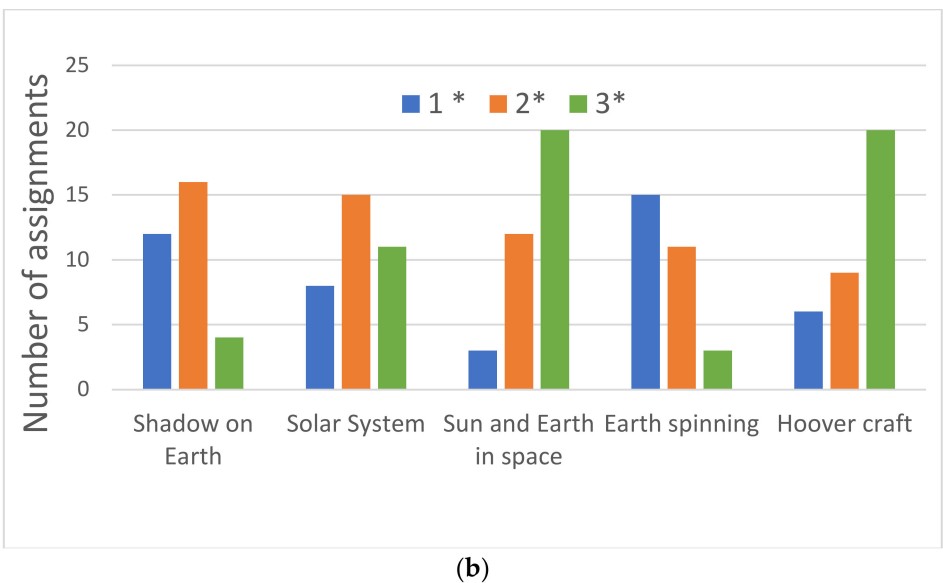

(**b**)

**Figure 15.** The results of the evaluation of the interactive plays on astronomy: "Which experiment did you like most? Put 1, 2 or 3 stars", NCU 22/02/2023. (**a**) Primary school (Dabrowa Biskupia, 20 students, 12 years old). (**b**) Secondary school (Torun, 34 students, 15 years old). The ordinate scale is the number of students who are assigned the given number of stars. The numbers have not been normalized, as the students were not expected to assign stars to every experiment. Lesson GK, 22/02/2023, evaluation by Katarzyna Wyborska, reproduced with permission.

## 4. Discussion

Our experience is that the use of didactic tools, relevant to the topic, but somewhat different in each lesson, creates interest in students: they wonder, "What interesting thing will happen in the next lesson?" This has been confirmed by one of our (ZhA) students, who after the role-playing game commented: "Your lesson is so interesting that it passes very quickly, we are waiting for the next physics lesson" and all the rest of the class agreed in unison. Another student said: "We forgot that physics was a very complex subject, now it seems to be just for us"; in the anonymous questionnaire we had another positive review: "When I come to the lesson of physics my mood rises; the teacher can joke with us, and we behave freely". For a teacher who is always searching for new, efficient and interesting methods, it is a great achievement and motivation for further work.

In the "duel game", the questions are not posed by the teacher, so they are a surprise. Only after looking at the questions prepared by students can the teacher learn the level of the student. Pupils themselves agree that during the preparation of questions they needed to understand the topic and find the subjects to repeat. This kind of game helps to repeat the theme, depending on the knowledge of the very subject, and also to learn arguments associated with the scenarios.

"The court of justice" is just a general scheme that can be proposed for different subjects. The question about electrical resistance, at first instance, seems not to be very theatrical but the same concept can be repeated in other themes, with different divisions of tasks. Important pedagogical aspects are the peaceful argumentation the children have to learn, the ability to assume a personal attitude, and the spirit of collaboration, at the end of the game. During such lessons and in the tournament where pupils change their roles in the game, they can form and develop further skills, which are shown in Figure 16.

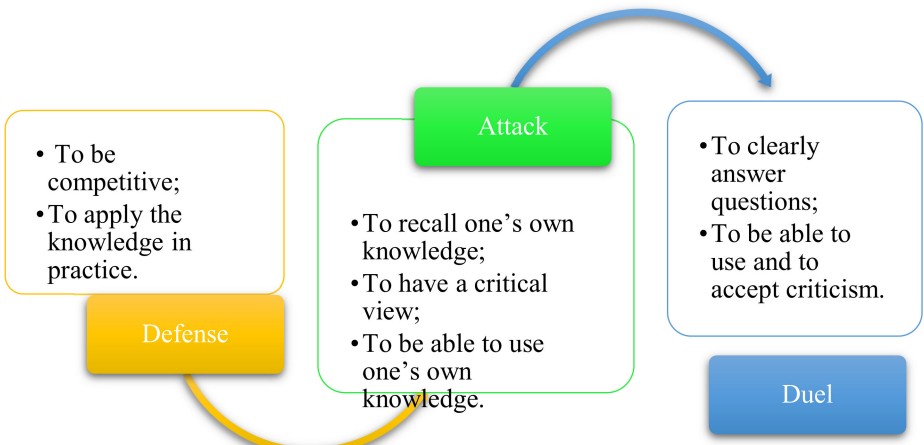

**Figure 16.** The skills that are formed during the role-playing game. Scheme by the author.

After the survey, students who did not use the game method expressed their desire to try it in physics lessons. They also said that they heard how other classes used the game and made the lessons interesting; this feedback once again proves that the use of the game in teaching physics really contributes to making learning easier and more interesting.

We have observed that the main effect of using games is better remembering of knowledge by a significant proportion of students. It is notable from the evaluation of the quiz that the used methods were developed with the help of students. The result can be considered a positive "mood" of students during the game, and their desire to learn. The use of competitive games in physics is particularly convenient during repetition or when fixing a chapter: we add some emotional involvement which backs up the mere, somewhat "sterile", notions.

Our results agree with those of similar role-playing games applied to other problems. Segoni [46], in Italy, on the subject of the environmental protection, reported that students

enjoyed the game (3 answers out of 16) or enjoyed it very much (13 answers) and found it useful (3 answers); this agrees in general with our questionnaire, reported in Figure 14 (and in the text, on the utility of the games in teaching different subjects).

In the subject of astronomy, Francis [45] reported the results of games performed in 30 schools all over the world. He quotes similar statements to those we heard from our students organizing the astronomical tunnel: "We enjoyed these tasks more because it gives us the sense that we're the first ones to discover these things [ . . . ]". This is exactly what happened in Almaty: our initiative was, certainly, not new on the world scale (as said, similar tunnels were organized at NCU in 2007), but everybody—the school authorities, teachers, visitors, and students organizing the interactive path had the impression of their own, authentic, personal discovery.

Note, however, that both NCU and KazNu are specific cases: they are leading national universities. Interactive exhibitions in Poland were started by one of us (GK) in 1997; the schools in Almaty where we performed the implementations are reserved for selected students: we have practiced different types of active games and plays in educational processes only there. So, the road to the full introduction of our scenarios to national educational systems is still long. Additional trials, in different environments, are needed.

As already stated, we report here the result of the first implementations of our didactical games, so we still need a more precise evaluation of the impact. This will be done in the next academic years, with a bigger number of schools involved, but already the present results prove that the design of the games is educatively promising.

The long-term impact of interactive role-playing and interactive didactical "tunnels" is difficult to quantify; by the very concept, these forms should more induce interest than perform comparative teaching (see [31] for the discussion on the equilibrium between the didactical and ludic functions of interactive didactics). An indirect measure is the spinoffs that our forms trigger. Lessons on interactive astronomy are performed every year at NCU, for the birthday of Copernicus. Similarly, experiments inspired by Galileo's work in physics and astronomy are part of many of our lessons. In Figure 17, we show two photos from our most recent (22–23 February 2023) rehearsals. Note that these lessons are not free of charge for pupils, and in spite of this the interest is steadily rising. When we started some 15 years ago, we performed these lectures once a year; now, to answer the demand, we need three several-day "slots" in different seasons of the school year.

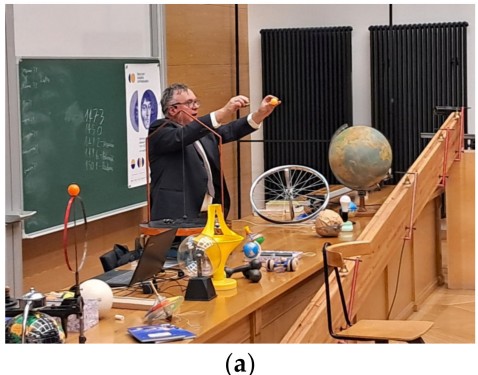 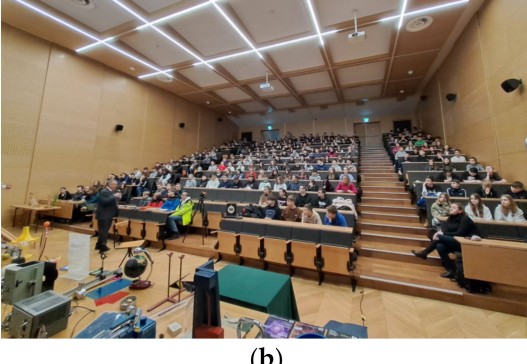

(**a**) (**b**)

**Figure 17.** The interest in interactive plays at NCU, in spite of the presence of many science centers in Poland, is steadily rising. In February 2023, in two days some 1600 students from elementary and secondary schools from the whole of Northern Poland came to Torun. (**a**) The key to the success is interactive experiments (Galileo inclined plane in the first plan, rotating chair, etc.). (**b**) The most difficult task is to keep the attention of the public for some 60–70 min, to interlace the interactive plays and run scientifically rigorous narration. Concepts and performance GK, photos Kamil Fedus, with permission.

## 5. Conclusions

The use of game technology is one of the methods that contribute to the comprehensive development of students. We discuss that in the age of the "virtual" world, coming back to face-to-face interaction in an educational game brings a lot of fun. The game is a process that does not depend on age: it involves, it cheers you up, it makes you think. From the pedagogical point of view, it is an efficient method to develop endurance, enterprise, attention, curiosity, multitasking, and other qualities.

Since in this work we conducted research on various age characteristics, we come to the following conclusion: with the help of a game one can teach both young children and teenagers even complex subjects, such as physics and astronomy, provided that the game is organized thoroughly and is related to the subject taught. Triggering pupils' interest and their autonomous will to study is an even more important result of interactive games during school lessons than the mere notions transmitted. Learning physics, in this way, is no longer "depressing" or "monotonous".

These results agree with those of other studies, like the comparison between online and on-site learning: innovative methods increase efficiency. Unfortunately, game techniques are not commonly used, especially in the teaching of physics in Kazakhstan or Poland: despite the role-playing methods in the classroom bringing much fun, they require safety and thorough organization. The new methods necessitate also a deep, interdisciplinary knowledge of the teacher, on both scientific subjects as well on pedagogical aspects.

Pedagogically, teaching methods through games and plays create an atmosphere of mutual cooperation and personal involvement. In addition, such didactics encourage schoolchildren to engage in independent interdisciplinary research work without giving priority to one narrow area, as we deduce from the academic careers of young "astronomers" at competitions held in Kazakhstan.

More extended implementations of the proposed methods will require complementary activities: presenting results in national journals on physics and astronomy education, at congresses and, primarily, testing in more schools and/or universities. All these activities are currently run by authors.

To conclude, the success of the game methods depends, obviously, on the individual capacities of the teacher to engage intellectually (and emotionally) the "players", and, in the age of the "virtual" world, games involving personal interactions are still attractive.

**Author Contributions:** Conceptualization, Z.A., K.T. and G.P.K.; Methodology, Z.A. and G.P.K.; Software, G.P.K.; Validation, Z.A. and G.P.K.; Investigation, Z.A., K.T. and G.P.K.; Data curation, Z.A.; Writing—original draft, Z.A., K.T. and G.P.K.; Writing—review & editing, G.P.K.; Supervision, K.T. and G.P.K.; Funding acquisition, G.P.K. All authors have read and agreed to the published version of the manuscript.

**Funding:** The PhD fellowship of ZhA is covered by Al-Farabi Kazakh National University. The fellowship of ZhA for the stay at Nicolaus Copernicus University has been covered by NAVA programme of Polish Ministry of Science and Education. Other costs, including the cost of publishing have been covered from funds of Didactics of Physics Division, NCU.

**Institutional Review Board Statement:** The study was conducted in accordance with the Declaration of Helsinki, and approved by the Ethics Committee of General Data Protection Regulation (protocol code EU 2016/679 and the date is 27 April 2016).

**Informed Consent Statement:** Informed consent was obtained from all subjects involved in the study.

**Data Availability Statement:** Additional materials of this research will given at Didactics of Physics Division, NCU, http://dydaktyka.fizyka.umk.pl/nowa_strona/?q=node/1012 (accessed on 1 April 2023).

**Conflicts of Interest:** The authors declare no conflict of interest.

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
