# Peer review of "Interactive Games and Plays in Teaching Physics and Astronomy"

_education, doi:10.3390/educsci13040393_

Round 1

Reviewer 1 Report

Review Report

Article title: Interactive games and plays in teaching physics and astronomy

The manuscript presents the results of an implementation of interactive plays and games for enhancing pupils’ interest and understanding of physics, astronomy, and engineering at schools in Almaty, the former capital city of Kazakhstan. The teaching materials were prepared in collaboration between two universities: the Nicolaus Copernicus University and the Al-Farabi Kazakh National University. The educators relied on two didactic principles in their activities: hyper-constructivism and neorealism. I highly appreciate the activity the authors have developed to increase the student’s interest in physics. However, it is necessary to make substantial changes to the structure and content of the submitted manuscript. The text in its present form does not have a character of an original research manuscript. It does not report scientific experiments.  I recommend the authors to revise the manuscript and change the article type to review (see instructions for authors available at https://www.mdpi.com/journal/education/instructions).

Specific comments:

If I were to review the manuscript as an original manuscript research, I must state the following:

1.      The research objective is not clearly stated in the abstract of the manuscript.

2.   In the abstract, the methodology used and the sample from which the data were obtained should be explained. Furthermore, the results should be summarized.

3.    The introduction should include the objective of the research. Furthermore, a state-of-the-art review of current scholarly literature on different approaches to physics teaching is necessary and should be included it in the paper.

4.   The text of the paper does not provide the exact wording of the research questions and the research hypotheses. The paper would be significantly improved with the inclusion of additional details about educational learning theories which the authors relied on.

5.      In the “Materials and methods” section the research sample should be characterized (number of students, age of students, school, etc.).

6.  The authors state in the text: “The didactical target is pupils of primary and secondary schools (aged 8-16), both in Poland and Kazakhstan.” The didactic goal is not properly formulated.

7.      What research methods and research tools did you use?

8.      How was the validity and reliability of the obtained data ensured?

9.      The obtained results are neither statistically processed nor evaluated.

10.  There is a mistake in sentence on line 216. It should read: How long does it take to Earth to make a complete ROTATION?

11.  The authors state in the text: “Questions that are supposed to be answered are given in Table 1.”  Table 1 does not exist. It has neither a caption, nor a header. From a didactic point of view, it is not correct to formulate sentences like "What is...?" in physics teaching.

12.  On page 12 there is again Table 1, which also does not have the correct form. On top of the table, the authors state: “Table 1. Comparison of quiz results before (left part) and after (right part) applying game teaching. Two experimental classes, in which the interactive role-playing game was used as a refresher lesson are compared with the control group (8B), in which consolidation was done in the traditional way. The first series of five columns give results before the consolidation lesson. The next five columns give results after the consolidation.” However, nowhere in the text is it stated who made up the experimental and control groups, how they were selected, etc. The text also lacks a detailed description of the activities of the experimental (EG) and control (CG) groups.

13.  In the graphs in Figs. 12 and 14 the description of the axes is missing.

14. The caption of Fig. 3 lacks the designation c). Similarly, in Figs. 5, 6, 7, the designation a), b), and c), which the authors refer to in the text, is missing. I recommend unifying the type of this marking in all images, since the authors sometimes use the marking a), b), and sometimes also a, b or A, B.

15.  The “Discussion” section should be enriched with the results of foreign research and compared with the results obtained by the authors of this article and of the working hypotheses. The findings and their implications should be discussed in the broadest possible context and the limitations of the work should be highlighted.

16.  All humans in the photographs must be blindfolded to ensure anonymity. Otherwise, a written consent from the person depicted is required.

17.  The section ‘Summary’ needs to be better unfolded, highlighting what is new in the research and how the results of the study can be used in practice.

Author Response

We thank the referee for valuable comments.

All suggestions have been included into the revised version.

Point-by-point answers (referring to lines of the revised manusrcipt) are given in the attached file. 

Reviewer 2 Report

the article has been well written by the researcher, but there are things that need to be improved such as

1. gap analysis needs to be sharpened, why games and play? what are the problems in the field?

2. Are there any research subjects used in this study?

3. the photo displayed in the article, please justify whether the person being photographed has permission to publish?

Author Response

The article has been well written by the researcher, but there are things that need to be improved such as

  1. gap analysis needs to be sharpened, why games and play? what are the problems in the field? 

We discuss it now in detail in the introduction: why this kind of new didactical solutions is needed. We show the rather disastrous results of the maturity exams in Poland in 2022 as the additional proof (new figure 1)

2. Are there any research subjects used in this study?

The subject of the study are five innovative didactical solutions, of our authorship (now better listed in lines 483-503, see also our answers to the first referee), implemented in schools in Poland and Kazakhstan, adn evaluate in a partel way). We stress that the paper is born to the special issue on designing educational games. We are aware that the experimental basis is still narrow. We describe our current actions to expand the applications of our proposed methods in the Discussion.

3. the photo displayed in the article, please justify whether the person being photographed has permission to publish?

Yes, obtained

English has been checked by the native speaker. 

Round 2

Reviewer 1 Report

Dear Authors,

I appreciate the great efforts you have made in response to my previous questions and comments. The revision clarifies almost all points I raised. You have significantly improved the clarity of your writing and addressed most of my concerns.

The text of the manuscript needs to carefully proofread to avoid misprints, e.g., line 34 “educational instruments, n that requires”, commas in sentences, periods at the end of sentences, etc.

In the sentence in line 99 you state, “In spite that these books were published by leading editors in science education, their influence on teachers' methods seems null.” it needs to be supported by a relevant citation of research results.

The research objective and research questions (or research hypotheses) should be listed at the end of the Introduction section, not in the Materials and Methods section (see instructions for authors available at https://www.mdpi.com/journal/education/instructions).

Given that you did not focus on evaluating the impact of the games you designed on students' academic results, I recommend reformulating research question no. 3.

Tables do not have the format specified in the Microsoft Word template.

Kind regards,

The Reviewer

Author Response

Thank you for the comments.

We did careful reading and remove miss-prints noted - some of them come from multiple editing with the Editor: the final check will be done in the accepted draft. 

>The research objective and research questions (or research hypotheses) should be listed at the end of the Introduction section, not in the Materials and Methods section (see instructions for authors available at https://www.mdpi.com/journal/education/instructions).

We added additional definition of objectives at the end of the introduction (lines 161-169), but we can not move the entire discussion on objectives from "Methods". As we already stressed, this paper is being submitted to the special issue on "Designing educational games": we must first explain what we designed, and only then which were the objectives related to each specific game that we have developed.

>Given that you did not focus on evaluating the impact of the games you designed on students' academic results, I recommend reformulating research question no. 3.

The game was applied to secondary school pupils, but not the whole classes - only to the best students. To our knowledge, all of them enrolled into high-level universities. This is the rough evaluation of the impact that we are able to give. 

Line 99 have been explained (new lines 101-102).

We leave the format of the tables to the pre-final editing by MDPI, according to our previous experience with this editor.

Sincerely yours

Authors